# A Patulous Progress: International Entrepreneurship Effects on Chinese Born-Global Firm Performance

**Ying Yu** [1,*] , **Xiaoling Hu** [2] , **Yong Wang** [3] **and Philippa Ward** [2]

1    School of Public Administration, Zhongnan University of Economics and Law, Wuhan 430072, China; yuying@zuel.edu.cn
2    School of Business and Technology, University of Gloucestershire, Cheltenham GL50 2RH, UK; xhu@glos.ac.uk (X.H.); pward@glos.ac.uk (P.W.)
3    Business School, University of Wolverhampton, Wolverhampton WV1 1LY, UK; yong.wang@wlv.ac.uk
*    Correspondence: yuying@zuel.edu.cn

**Abstract:** Using data from SMEs in Hubei province, the role of entrepreneurship in the sustainable performance of born global firms in China was discussed. The structural equation modeling analysis of 345 questionnaires indicates that both international knowledge and international entrepreneurial capability are significantly related to born global firms' performance. Dynamic capabilities, which includes three sub-dimensions: adaptation capability, absorption capability and innovation capability, was found to be less important to firm performance. Therefore, it may be argued that born global firms in inland China are still limited by resources, including those generated from the international knowledge needed to adapt to internal and external pressures.

**Keywords:** internationalization; small and medium sized enterprise; born-global firms; international entrepreneurial capability; dynamic capabilities; international knowledge

## 1. Introduction

In developed and developing countries, small and medium sized enterprises (SMEs) are the backbone of the economy [1]. They not only contribute to a country's exports [2] but also play a significant role in employment generation [3]. In emerging economies particularly, SMEs that experience rapid economic growth, with increasing income and buying power, create significant economic and social development impact [4]. Hence, SMEs in emerging economies are an engine of growth and worthy of examination.

Unsurprisingly, there are growing efforts to research SME internationalization and specifically international entrepreneurship's role in China. However, better contextualization is needed in this examination [5,6]. This research seeks to further advance such theoretical understandings of SMEs and international entrepreneurship within China.

Conventionally, the Uppsala model [7] is used to understand SME internationalization as the the most commonly adopted internationalization market entry strategy. It suggests firms initially gain a domestic base, then gradually expand internationally to mitigate embedded process risks [8]. Firms which adopt the Uppsala model seek to establish a solid domestic base at first, and then gradually expand into international markets, with the justification for this relating to the risk and uncertainty embedded in the internationalization process [8]. One of the major characteristics of this type of entry mode is that firms' internationalisation tend to be slow, gradual and incremental. This conceptualization advocates that internationalization only occurs once a firm is established domestically and thus well after its start-up. However, firms engaging in international business early in their development are emerging in significant numbers [9]. In the last two decades, increasing

numbers have rapidly entered international markets after their inception [10]. This phenomenon challenges the Uppsala, or 'stage' model and has therefore attracted considerable research [11,12] that contends not all firms pass through the same 'international' export development cycle.

Whilst there is agreement that some businesses, from inception, engage in international markets; varied terminology is used to describe such firms including: "born-globals" [13,14], "global start-ups" [15], "high technology start-ups" [16] and "international new ventures" [17]. Irrespective of the nomenclature, such firms subvert the Uppsala model. This is evident when considering a 'born-global', defined as: a firm that seeks significant international competitive advantage from inception and operates and uses resources in multiple countries within three years of establishment [17,18], additionally it derives at least 25% of turnover from international business [19]. To achieve such outcomes, born-globals generally display entrepreneurial mind-set and risk tolerant behaviour [20]. At the same time, the emergence of digital platforms, innovation platforms and ecosystems enables new ways of creating and delivering value to global customers, thus come to dominate many industries' landscape and shape the ecosystems to their advantage [21,22]. Globally, the organizational unlearning from young and aggressive multinational corporations becomes a means for inducing multifaceted learning that enables these companies to develop and exploit their entrepreneurial capabilities [23].

The existing literature on born-globals emphasizes the importance of 'international entrepreneurship' [2]. International entrepreneurship was defined by McDougall [15] as "a combination of innovative, proactive, and risk-seeking behavior that crosses national borders and is intended to create value in organizations". Furthermore, entrepreneurship-related factors are more important than external dynamics for born-globals, as they need rapidly to exploit entrepreneurial opportunities [24]. This is because the entrepreneur is at the core of a born-global [25]. These entrepreneurs are responsible for all decisions and numerous studies demonstrate that their 'international entrepreneurial capability' and 'dynamic capabilities' [26] are central to firm success [27,28].

However, research on Chinese born-globals that considers the relationship to international entrepreneurial capability and dynamic capabilities is limited [29]. Since international entrepreneurship is a multi-faced concept, we attempt to take a more holistic view, considering factors including adaptation- and absorption capability to enhance the international entrepreneurship literature and connect it more firmly with firm performance (viewed as 'sales', given the importance of international sources of turnover to the 'born-global' definition). Thus, the research questions are: (1) what are the chief underlying factors determining international entrepreneurship for Chinese born-globals? (2) what is the relationship between international entrepreneurship and business performance in Chinese born-globals? To answer these questions, a first and second order structural equation model is constructed and tested.

## 2. Theoretical Underpinnings

Our aim is to investigate whether the international entrepreneurship displayed by those managing Chinese SMEs affects firm performance in a global market. International entrepreneurship is a multi-faced construct and interrelated with many other factors, including networking capability [9], dynamic capabilities [30,31]. Therefore, a holistic view is adopted including these aspects. The principal factors pertinent to born-globals included are dynamic capabilities and international knowledge and a rationale for their selection is presented below.

### 2.1. Dynamic Capabilities Theory and Born-Global Performance

Dynamic capabilities are defined as "a firm's behavioural orientation constantly to integrate, reconfigure, renew and recreate its resources and capabilities and… upgrade and reconstruct its core capabilities in response to the changing environment to attain and sustain competitive advantage" [26]. Dynamic capability theory highlights the ability that firms possess that enables them to achieve

organizational or strategic goals. Such an ability may not only help firms to obtain new resources when they experience events such as market emergence or evolution, but could support them to appropriately adapt, integrate, and reconfigure these resources as market conditions vary [26,32]. Zhang [32] believes that effectively allocating capabilities could help firms gain a competitive advantage in their internationalization process. Hence, dynamic capabilities are linked with firm performance as they efficiently allocate and maximize resources and capabilities, then influencing firm performance [31].

The process of building dynamic capabilities involves knowledge creation, integration and configuration [33]. According to Weerawardena [34], the process of capability building in born-globals is driven, and accelerated, by entrepreneurial owner-managers with a global mind-set, prior international experience and a learning orientation. Such individuals build and nurture the distinctive capabilities that enable the small, international-oriented new ventures to develop niche knowledge-intensive products. Similarly, Eisenhardt [35] also stressed that dynamic capabilities enable firms to transform and apply resources and capabilities into outputs such as goods or services sooner than the other firms. Dynamic capabilities also enable a firm to renew and recreate its resources and capabilities [31]. Namely, a firm that possesses dynamic capabilities can enjoy a transformational competence which, via learning and absorbing, transforms market information efficiently into appropriate adaptation, and eventually supports new products or service development through innovative behaviours [36,37]. Given the centrality of dynamic capabilities to innovation, and the role this plays in a firm's strategic development, examining the dimensions of dynamic capabilities is critical. According to previous studies [26], the three dimensions of dynamic capabilities investigated in this study are listed below.

- `Adaptation capability`

Adaptation describes the ability to be strategically flexible so that firms can adopt emergency plans, when necessary, to meet new tendencies emerging in their environment [38]. This capability is essential as it determines the adaptation of current resources based on knowledge management skills [39]. Firms with the capability to manage their knowledge base can effectively reduce response times when experimenting, or explicitly implementing, new techniques [40]. To develop adaptation capability to facilitate knowledge integration and knowledge transfer, firms are also required to possess effective management and a positive attitude towards this process [41]. A firm operating in international markets quickly after inception is likely to encounter emerging new tendencies and examining its capability to adapt to these by marshalling its knowledge management skills offers potential insight into a born-global's dynamic capabilities.

- `Absorption capability`

The second dimension of dynamic capabilities investigated is a firm's absorption capability. Building absorption capability relies on the knowledge a firm possesses and the ability to transfer knowledge or apply it to a firm's products, processes or personnel [42,43]. A higher level of absorption capability is believed to enable firms to possess a stronger ability to learn from other firms, then assimilate this external knowledge and transfer it into their own internal knowledge base, eventually successfully applying its own business activities [44]. Geirge [45] suggests that a firm's ability to acquire external knowledge, then absorb and integrate it with existing (internal) knowledge to create new knowledge is essential in developing dynamic capabilities. As a born-global will encounter other companies across international markets, the opportunities to learn from diverse firms may be enhanced. To capitalize on this, a born-global would need to display higher levels of absorption capability to improve its overall dynamic capabilities.

- `Innovation capability`

The third dynamic capabilities dimension is innovation capability. It describes a firm's ability to create innovative ideas, new products or new processes to meet market demand [46]. With

increasing competition in contemporary markets, innovation capability facilitates the gain, exchange, gathering, integration and development of valuable knowledge and resources from individual agents by formulating inter-organizational processes and routines [47]. This should be of importance to a born-global and this capability would significantly increase the firm's dynamic capabilities.

It is also argued that dynamic capabilities are used extensively in the development of new products that then support the change and renewal of firms [48]. Furthermore, these dynamic capabilities also encourage firms to be more open to innovative ideas and establish innovation as a part of their culture, which eventually benefits their innovation capability development [49].

Given this discussion, hypotheses related to the relationship between dynamic capabilities and the sales performance of born-globals can be formulated:

**Hypothesis 1:** *The sales performance of born-globals is significantly influenced by their dynamic capabilities.*

**Hypothesis 2:** *The higher the level of adaptation capability, the more dynamic capabilities born-globals will possess.*

**Hypothesis 3:** *The higher the level of absorption capability, the more dynamic capabilities the born-globals will possess.*

**Hypothesis 4:** *The higher the level of innovation capability, the more dynamic capabilities the born-globals will possess.*

*2.2. International Entrepreneurship Capability and Born-Global Performance*

Internationalization occurs because entrepreneurs see opportunities in foreign markets that act as "open windows" for their firms. For the internationalization process to succeed, entrepreneurs must strengthen their entrepreneurial skills, such as in risk taking, creating new products, establishing network contacts, and utilizing special knowledge gained via different channels [2]. Therefore, the role of entrepreneurship in internationalization is well recognized [50]. International entrepreneurship is hence a process that integrates the entrepreneur's ability to forecast the result of actions and the realization of business emergence as an international entity [2]. Within the realization of international entrepreneurship, international entrepreneurial capability is defined as a firm's leveraging of resources by combining a series of innovative, risky and proactive activities [32]. International entrepreneurial capability also enables entrepreneurs to explore, enact, evaluate, and exploit opportunities internationally. In this study, we use 'international entrepreneurship' and 'international entrepreneurial capability' interchangeably.

Entrepreneurial behaviour has been found to be a significant common denominator in born-global studies (e.g., Svante [51]). We thus, believe that international entrepreneurial capability has a direct bearing on born-globals' sales performance. International entrepreneurial capability emphasizes the activities related with brokering, resource leveraging or stretching, value creation and opportunity seeking [32]. The success of these activities requires firms to adopt innovative, proactive and risk seeking behaviours [15]. Indeed, Zhang [32] claimed this international entrepreneurial capability is especially helpful for born-globals because they often lack financial and human capital, along with tangible resources. Thus, international entrepreneurial capability can help born-globals achieve superior performance by leveraging their limited resources. International entrepreneurial capability is clearly of importance for any SME, but its role and prominence are more sharply and swiftly brought into focus for a born-global. In such a firm, the entrepreneur must be adept within the international arena to propel the firm and hence considering what is required to do this in more detail is of critical importance.

To do so, the three key international entrepreneurial capability dimensions proposed by Zhang [32] are used. They are capabilities allied to: international networking, marketing and risk-taking.

- *International networking capability*

It was found that to accelerate the internationalization process, born-global firms usually adopt advanced communication technology to acquire knowledge, develop strategies and maintain relationships [52]. Here, international networking capability refers to the ability of a company to obtain resources from the environment through the creation of alliances and social bonds for use in their international market activities [31]. Whilst international networking capability may resonate with the notion of 'absorption capability', the focus is on the generation, and use, of the network (external) whilst that of absorption capability examines the firm's ability to integrate that information into the organization (internal). Building social networks allows firms to gain knowledge from other agents' behaviour, other firms' pricing or technology, and from observing collective actions.

Knight [52] noted that for SMEs, information technology and relationships are the two main tools deployed to deal with the uncertainties embedded in foreign markets. To extend their boundaries, firms use telecommunications and computer technologies to manage business ties with customers and suppliers [53]. Once a firm's networking capability generates knowledge about markets and technologies, it has been demonstrated to directly influence firm performance [54].

- *Marketing capability*

Blesa [55] claimed that marketing capability is a firm-specific ability that facilitates, or forms, success in international markets because this capability provides the firm with a superior market sense, customer link, and channel bond. Song [56] suggested that market capability is comprised of varied knowledge and information on issues including competition, customers, market segmenting, pricing, advertising, market integration activity. Moreover, Morgan [57] highlights that market capability is about how market strategy is developed and executed. Thus, in general, marketing capability is a firm's ability to use competition knowledge to develop and execute a market strategy to achieve superior performance abroad [52]. As such, it provides the foundation for firms to interact with international markets [52]. Likewise, firms with a developed marketing capability can analyse and understand customers' demands better, which eventually could facilitate the forging of new market segments and enhance firm performance [58].

- *Risk-taking capability*

Risk-taking capability refers to the ability to make decisions or resource commitments that contain substantial risk in foreign markets [32]. Internationalization is as a risk-taking behaviour for entrepreneurial firms, as not only are foreign markets full of uncertainties, but also such behaviour may cause serious funding problems (such as debt) in the opportunity exploitation process [46]. Given the discussion above, the following hypotheses are formulated:

**Hypothesis 5:** *The sales performance of born-globals is significantly influenced by their international entrepreneurial capability.*

**Hypothesis 6:** *The higher the level of networking capability, the higher the level of born-globals' international entrepreneurial capability.*

**Hypothesis 7:** *The higher the level of marketing capability, the higher the level of born-globals' international entrepreneurial capability.*

**Hypothesis 8:** *The higher the level of risk-taking capability, the higher the level of born-globals' international entrepreneurial capability.*

*2.3. International Knowledge and Born-Global Performance*

It is evident that a firm's dynamic capabilities have been associated with its performance, but if such organizational and strategic routines to achieve new resource configurations are important, so too are the resources that can be marshalled. As with other authors examining born-globals [59]

or entrepreneurial orientation [60], this research not only examines dynamic capabilities, but also consider the resources deployed and specifically international knowledge, which can be associated with building international networks and markets. The knowledge-based view emerged from the resource-based view [61]. It focuses on intangible resources rather than on physical assets. Scholars claim that the knowledge-based view positions organizations as accumulators of knowledge and competencies [62,63].

This theory has been extensively examined in connection with born-global internationalization (e.g., Johanson and Knight [52,64]). From this perspective, knowledge is the most important resource. Furthermore, the main cause for performance differences across firms are polyphase knowledge bases [65], which means "the development, integration, and transfer of knowledge should be regarded as a critical aspect of strategic management of internationalisation" [64]. Through experience-based learning in non-domestic environments, SMEs can generate 'experimental' knowledge and adopt this knowledge across foreign markets [66]. Researchers also acknowledge that new international ventures, or born-globals, can accumulate and transfer knowledge faster than other firm types [67]. Miller [68] believed that in dynamic environments, born-globals that operate knowledge-based resources can gain better performance than firms operating property-based resources.

Based on the knowledge-based view, Tsinopoulos [69] stressed that international knowledge is an intangible asset and provides a competitive advantage for firms in foreign markets. Zhang [70] also claimed that both individuals and firms can utilize existing knowledge resource from prior cross-border business experiences to improve their learning from the foreign market entry experience. For a successful entrepreneur, these experiences are precious and retrievable, which includes how to read cultural cues, how to build trust, how to identify business opportunities, how to negotiate international contracts and so on [66].

It is acknowledged that entrepreneurial firms that possess more international knowledge can become accustomed to foreign markets [13]. Some of the differences between the traditional internationalized firms and born-globals can be explained by the founders' overseas living, working or studying experiences [13]. Entrepreneurs who have previous practical business experience either from working in a commercial environment or starting a business in a foreign market, have comparative advantages compared with others who do not [71]. Additionally, research shows companies with a manager who lived or studied abroad are more likely to engage in international activities [72].

Thus, for born-globals, international experiences are essential in their market entry process. In this study, internationalization knowledge (as a resource) is considered as the experience of having conducted international business, such as foreign direct investment (FDI), prior experience with foreign partners or received education abroad [73].

Given the discussion above, the following hypothesis is formulated:

**Hypothesis 9:** *The firm performance (sales) of born-globals is significantly influenced by the owner/manager's level of international knowledge.*

## 3. Methodology

### 3.1. Sampling and Procedures

Primary data were collected from Hubei province in central China.

It should be acknowledged that due to the diversity and immensity of China, it is impossible to identify a representative sample. The previous studies on born global firms collected data from one or two regions in China [32,74], so as in this study. The location of our primary data collection is focused on firms located in Hubei province in central China. Nevertheless, it may be acknowledged that the limitation of choosing specific region to collect data sample compromise the generalizability of the study, but it is believed the region that the data is collected for this study is quite representative among the most of other parts in the country. Hubei's total GDP ranked 7th in China with a total GDP of RMB

4582 billion (2019). The annual GDP growth rate in 2019 was 7.5% well above the national average 6.1%. Wuhan (the capital of Hubei Province), the biggest city in central China, is the focal point of the "Rising of Central Regions Strategy". This strategy is initiated and undertaken by the Chinese government to develop the central regions of China economically, which has already experienced a significant industrialization and economic growth in recent years. According to the view of Culture researchers [75], the transformation of a region from under developed area to develped area will change people's values and cultures, and propels they converge towards to the values and cultures in developed regions. Therefore, peoples in Hubei province are increasingly similar in values and cultures to the Eastern coast regions for the capacity relocation.

The region was chosen for during the outbreak of Covid-19, this province was almost locked down for two months and the enterprises there faced with severe difficulties. As Cardoza [74] states, choosing a specific region within which to collect data may compromise the study's generalizability. However, we believe region's culture should be quite representative among most other areas of China, which may lessen generalizability problems. Wuhan, the municipal city of Hubei Province is a special place in China for it is a mixture of multiple missions both historically and economically, it served as the capital of the nation, a heavy industry base, one of the first ports opened to the world with a special court inland but hears the international conflict cases. The Covid-19 outbreak makes it global again. Culture researchers argue that a region that develops rapidly changes people's values and cultures and will increasingly converge towards the culture and value systems of the more industrialized regions [75]. Thus, it is rational to conclude that people in Hubei province will be increasingly similar in terms of their values to those in China's developed regions. Additionally, as most areas in China are still experiencing development, focusing on a developing region such as Hubei may be more representative.

A questionnaire was designed in Chinese (based on the extant literature) and given to target firms. A seven-point Likert scale was used to obtain respondents' answers. Each question is an attitudinal statement measured through a 'strongly disagree' to 'strongly agree' scale.

Respondents were middle- or high-level managers, directors, or project managers of small and medium-sized firms involved with international activities. We believe these respondents not only possess sufficient knowledge of management issues and firm performance, but also a degree of involvement with the firm's decision-making processes.

SMEs in Hubei with foreign sales were randomly selected from the list of export firms compiled by the Custom Bureau of China. In total, 1000 questionnaires were sent by a professional research agency, 345 complete questionnaires were received, giving an effective response rate of 34.5%. Of these responses 172 fit the born-global criteria while the rest could be classed as traditional exporters. The definition used to identify the born-global firms is those that start international activities within three years of inception and achieve a minimum 10% of export as a percentage of total sales. The data in this paper were collected from March to July in 2017, an approximate end of China's high speed railway network and the economic geographic landscape of central China did not change since then. Since then the Hubei province can be seen as the absolute central part of China in economic geography and its municipal city Wuhan is the only key node in the Belt and Road Strategy among the several provinces in central China.

Among the 172 born-global firms, According to the results, 54% are limited liability firms, 34.9% are private firms, 9.9% are joint stock limited liability firms, and only 1.2% are foreign funded firms. 117 respondents achieved export intensity equal to or higher than 20% but lower than 30%, 23 respondents achieved export intensity equal to or higher than 10%, but less than 20%. According to the results, among 172 respondents, 54% are limited liability firms, 34.9% are private firms, 9.9% are joint stock limited liability firms, and only 1.2% are foreign funded firms. In total, 117 respondents achieved export intensity equal to or higher than 20% but lower than 30%, 23 respondents achieved export intensity equal to or higher than 10%, but less than 20%. 27.9% of them were from manufacturing, 20.9% of them were from construction, 18.0% of them were from tourism, 16.7% of them were from wholesale and retail trade, 9.9% of them were from electricity, gas and water supply, 3.5% of them

were from pharmaceutical industry, 1.7% of them were from transport, storage and communication. Among the 172 managers, 76.1% are male and 23.9% are female, 79.6% received undergraduate level education, 12.2% received graduate level education and the remaining 8.2% received other types of education. 36.3% of the managers were at their 20s, 41.9% were at their 30s, 17.4% were at their 40s and 4.1% were managing the firm with an age above 50.

### 3.2. Constructs and Variables

Based on the extant literature and previous empirical studies, a structural model was developed, see Figure 1.

As an exogenous variable, 'firm sales'—which can be viewed as an indicator of competitive advantage—is affected by two second order constructs and one first order construct. The second order constructs include 'dynamic capabilities' and 'international entrepreneurial capability'. The first order construct is 'international knowledge'. All are endogenous variables that have been shown as having positive impacts on business (sales) performance in previous studies [13,26,32]. The second order construct 'Dynamic Capabilities' is measured by three first order constructs: 'Adaptation Capability', 'Absorption capability', and 'Innovation Capability'. The second order construct 'International Entrepreneurial Capability' is measured by three first order latent variables 'International Networking Capability', 'Marketing Capability', and 'Risk Taking Capability'. 'International knowledge' is measured by foreign business experience and foreign institutional experience and finally, firm performance is measured by sales. This model is tested using AMOS 24 software.

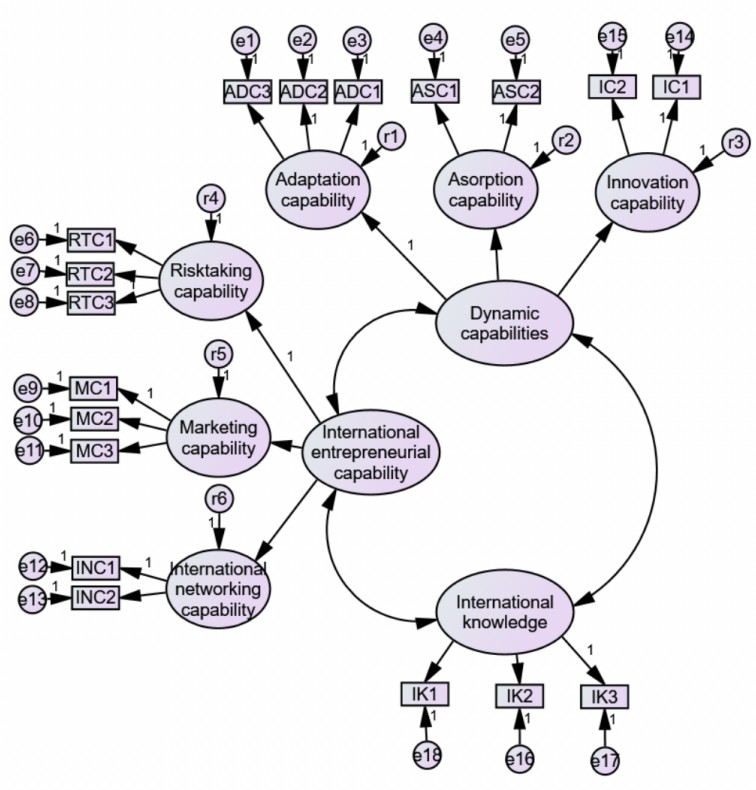

Chi-Square=\cmin; Prob=\p; GFI=\gfi; AGFI=\agfi
;CFI=\cfi; TLI=\tli; RMSEA=\rmsea

**Figure 1.** Measurement model for international entrepreneurship.

*3.3. Data Analysis*

- *Factor analysis*

The Harman one-factor test was conducted by Podsakoff [76] with seven first order constructs including absorption capability, adaptation capability, innovation capability, risk taking capability, marketing capability, international networking capability and international knowledge. The unrotated factor analysis showed no single factor arose and no dominant factor emerged to explain most of the variance. This implies that common method bias is not a significant issue in the current study.

Based on the literature review, 18 factors were identified. According to Howell [77] items can remain in a factor if they have load at, or above, 0.4, and the differences between this loading and two other cross-loadings are more than 0.3 [37]. Based on these criteria, 2 indicators were discarded, leaving 16 indicators for factor analysis. The factor analysis results show a KMO value of 0.881, which indicates the sampling is adequate; the chi square of Bartlett's test is 970.417, which is significant at 1% level with a degree of freedom of 120. The total variance explained is 59.56%. The remaining indicators were reduced to 7 dimensions. The names of these dimensions are listed in Table 1. The Cronbach's alpha coefficients range from 0.628 to 0.747, indicating the level of internal consistency and external independence are acceptable [78]. The dimensions and indicators for measuring born-globals' international entrepreneurship are also listed in Table 1.

- *Unidimensionality and convergent validity*

Unidimensionality means one latent trait or construct exists that underlies a set of indicators [79]. Here, we discarded 2 indicators (ADC3 and MC2), thus there are 16 items loading on seven (first-order) constructs. Table 1 presents the measurement properties of the constructs included in the second-order structure model of born-global international entrepreneurship. All items have significant loadings on their corresponding factors, evidencing good convergent validity and all structural coefficients show significantly high t-values.

The relative $\chi^2$, namely $\chi^2$/DF is 1.609. According to Byrne [80] a good fit for the observed data sample requires a s low $\chi^2$ value about its degree of freedom. Specifically, a relative chi-square value of 2 or less indicates a reasonably good fit for the model. In addition, the Comparative Fit Index (CFI) [81] for the model is 0.935, which is greater than the 0.9 recommended critical value. Finally, the root-mean-square error of approximation (RMSEA = 0.060) is also within acceptable ranges, indicating the model accounts for a substantial amount of variance [82].

The convergent validity test result is determined by the indicator's coefficient value and its standard error (SE). According to Gerbing [83], if the ratio between the coefficient value of the indicator and its standard error is higher than 2, this means the indicator dimension is significantly convergent, thus is valid for measuring its construct. In this study, all the CR value/SE values are greater than 2. These results imply that all indicators measure their constructs.

- *Discriminant validity test*

According to Anderson [84] the discriminant validity test measures the extent of differences between model dimensions to confirm whether the model is unique (or not). Thus, this study employed a chi-square test to examine the differences between the constrained model (i.e., where the correlation is fixed to 1) and the unconstrained model (i.e., where the correlation is released). Table 2 presents the Chi-square test results.

The difference between the values of chi-square in the two models is 39.515, which is larger than the critical value of 9.488. Thus, the null hypothesis that the constrained model fits better is rejected, and the unconstrained measurement model is more appropriate. Meanwhile, the chi-square value is lower in the unconstrained model compared with the constrained model, hence all model constructs are not perfectly correlated (Bogazzi and Philips, 1991), implying each construct is unique and independent.

**Table 1.** The internal consistency and convergent validity of dimensions and indicatiors.

| Constructs | Items | Previous scales | Weight | S.E. | C.R. | P |
|---|---|---|---|---|---|---|
| Dynamic capabilities | | Adapted from Wang and Ahmed (2007) | | | | |
| Adaptation capabilities | | | 1.000 | | | |
| | ADC1: Our firm is able to price products effectively according to the changes in the market. | | 1.000 | | | |
| | ADC2: Our firm is able to develop flexible processes to respond rapidly to changes and opportunities identified in the markets. | | 0.734 | 0.080 | 9.180 | *** |
| Absorption capability (ASC) | | Adapted from Wang and Ahmed (2007) | 0.859 | 0.101 | 8.460 | *** |
| | ASC1: Our firm is able to develop new product or modify existing products by acquiring information from competitors. | | 0.858 | 0.104 | 8.269 | *** |
| | ASC2: Our firm is able to learn, analyze and interpret useful information from the market. | | 1.000 | | | |
| Innovation capability (IC) | | Adapted from Wang and Ahmed (2007) | 0.299 | 0.079 | 3.783 | *** |
| | IC1: Our firm is committed to innovation and product development. | | 1.000 | | | |
| | IC2: Our firm has the ability to innovate by using knowledge from various sources to develop products efficiently and rapidly. | | 0.987 | 0.264 | 3.739 | *** |
| International entrepreneurial capability | | Adapted from Zhang et al. (2009), Zahara et al. (2000) | | | | |
| Risk taking capability (RTC) | | | 1.000 | | | |
| | RTC1: Compare to other firms, our firm inclines to take on projects with high risks. | | 0.825 | 0.111 | 7.398 | *** |
| | RTC2: Compare to other firms, our firm inclines to take on projects with high risks. | | 0.663 | 0.098 | 6.766 | *** |
| | RTC3: Compare to other firms, our firm are more prepared to meet new challenges. | | 1.000 | | | |
| Marketing capability(MC) | | Adapted from Zhang et al. (2009) | 0.909 | 0.143 | 6.354 | *** |
| | MC1: Compare to other firms, our firm is better at controlling and evaluating marketing activities. | | 1.000 | | | |
| | MC3: Compare to other firms, our firm is better at differentiating products based on the knowledge and marketing tools. | | 0.926 | 0.143 | 6.478 | *** |
| International networking capability (INC) | | Adapted from Zhang et al. (2009) | 0.945 | 0.136 | 6.934 | *** |
| | INC1: Our firm has the technology-based link with customers and competitors. | | 0.993 | 0.155 | 6.431 | *** |
| | INC2: Our firm has entrepreneurial collaborations with external partners. | | 1.000 | 0.136 | | |
| International knowledge | | Adapted from Zhang et al. (2009), Madsen and Servais (1997) | | | | |
| | IK1: Top management in our firm continuously communicates its mission to succeed in international markets to firm employees. | | 0.803 | 0.119 | 6.743 | *** |
| | IK2: Top management has sufficient experience in foreign direct investment (FDI). | | 0.918 | 0.135 | 6.797 | *** |
| | IK3: Our managers has sufficient language and foreign laws/norms/standards knowledge. | | 1.000 | | | |

$\chi^2(95) = 152.837$; CFI = 0.935; GFI = 0.900; RMSEA = 0.060; * $p < 0.10$; ** $p < 0.05$; *** $p < 0.01$.

<div align="center">**Table 2.** Δchi-square test.</div>

| Unconstrained Model | | Constrained Model | | Δchi-Square Test |
|---|---|---|---|---|
| Chi-square | DF | Chi-square | DF | 39.515 |
| 192.352 | 99 | 152.837 | 95 | |

## 4. Empirical Results

To test the hypotheses, we formulated the following regression equation model:

$$Firmsale = \beta_1 DC + \beta_2 IEC + \beta_3 IK + \delta$$

where

DC: Dynamic capabilities
IEC: International entrepreneurial capability
IK: International knowledge
$\beta$ : Regression weight
$\delta$ : Disturbance
Dimensions of second-order constructs are listed below:

$DC = \gamma_1 ADC + \gamma_2 ASC + \gamma_3 IC + \varepsilon_1$
$IEC = \gamma_4 RTC + \gamma_5 MC + \gamma_6 IINC + \varepsilon_2$
$ADC = \lambda_1 ADC1 + \lambda_2 ADC2 + \mu_1$
$ASC = \lambda_3 ASC1 + \lambda_4 ADC2 + \mu_2$
$ASC = \lambda_5 ASC1 + \lambda_6 ADC2 + \mu_3$
$RTC = \lambda_7 RTC1 + \lambda_8 RTC2 + \lambda_9 RTC3 + \mu_4$
$MC = \lambda_{10} MC1 + \lambda_{11} MC3 + \mu_5$
$INC = \lambda_{10} INC1 + \lambda_{11} INC2 + \mu_6$
$IK = \gamma_6 IK1 + \gamma_7 IK2 + \gamma_8 IIK3 + \varepsilon_3$

where

ADC: Adaptation capability
ASC: Absorption capability
IC: Innovation capability
RTC: Risk-taking capability
MC: Marketing capability
INC: International networking capability
FBE: Foreign business experience
FAE: Foreign academic experience
$\gamma$: Loading factor
$\gamma$: Loading factor
$\varepsilon$: Error term
$\mu$: Error term

- *Results for the main hypotheses*

Both Figure 2 and Table 3 shows that the H1 to H4 were supported as the expected positive sign is achieved. This implies that born-globals' sales performance is influenced by their dynamic capabilities to a degree. However, H5 to H8 were not supported, since international entrepreneurship capability is significant at 5% level with a negative sign. This is a surprising result, and possible reasons for this are explored in the discussion. Finally, H9 was confirmed (at 5%), this implies that born-globals' sales performance is influenced by international knowledge.

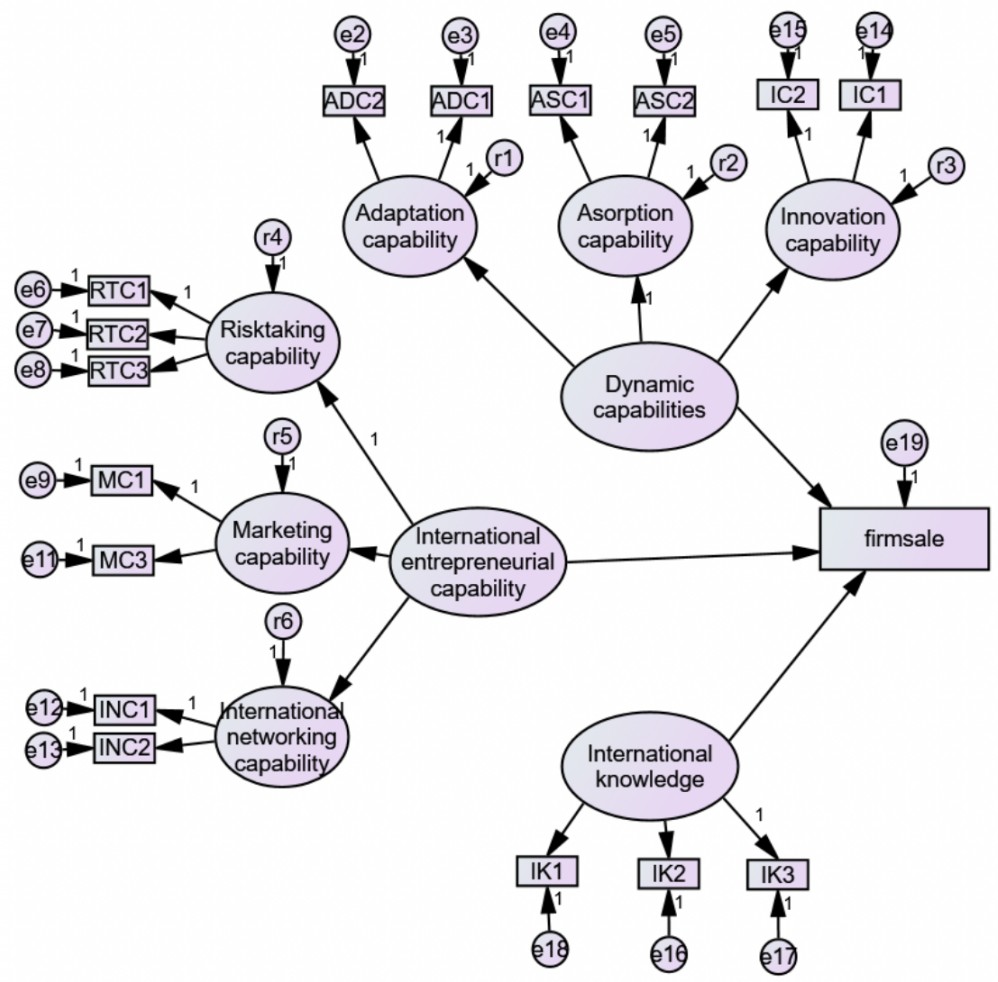

**Figure 2.** Structural equation model.

**Table 3.** Effects of entrepreneurship on performance.

| Item | Coefficient | S.E. | C.R. |
|---|---|---|---|
| Dynamic capabilities → firm performance | 3.746 | 3.000 | 1.249 (n.s.) |
| International entrepreneurial capability → firm performance | −7.889 | 3.715 | −2.124 ** |
| International knowledge → firm performance | 3.796 | 1.909 | 1.988 ** |

* $p < 0.10$; ** $p < 0.05$; *** $p < 0.01$.

- *Sub-hypotheses results*

The test results from evaluating the relationships between first-order and second-order constructs are significant, with *p* values lower than 0.001 (see Table 4). This suggests that all the sub-hypotheses can be accepted, implying the second order constructs have been appropriately depicted by their first order constructs. For instance, H3 is supported given the significant results (at 5% level), which implies the higher the level of absorption capability, the more dynamic capabilities born-globals possess.

**Table 4.** Effects of entrepreneurship on performance.

| Item | Coefficient | S.E. | C.R. |
|---|---|---|---|
| Dynamic capabilities → Adaptation capability | 1.000 | | |
| International entrepreneurial capability → Firm performance | 0.884 | 3.715 | −2.124 ** |
| International knowledge → firm performance | 0.520 | 0.088 | 5.909 ** |
| International entrepreneurial capability → Risk taking capability | 1.000 | | |
| International entrepreneurial capability → Marketing capability | 0.949 | 0.131 | 7.269 ** |
| International entrepreneurial capability → International networking capability | 0.950 | 0.144 | 6.602 ** |

* $p < 0.10$; ** $p < 0.05$; *** $p < 0.01$.

## 5. Discussion

The results show that entrepreneurs in Chinese born-globals see international knowledge as related to firm performance. This implies that born-globals that possess more international knowledge are more likely to perform better. The relationship between dynamic capabilities and firm performance achieved the expected sign but was not significant, while the relationship between international entrepreneurial capability and firm performance surprisingly is negative. Some of our results are inconsistent with previous studies where both dynamic capabilities and international entrepreneurship are found to play particularly important roles in firm performance [26,32]. The insignificant, or even negative, results here may be accounted for in a variety of ways. First, the international entrepreneurial capability result reveals the difference between the entrepreneurs' perception of their international entrepreneurial capability and their firms' actual performance. This indicates that, in the short-run, the development of international entrepreneurship was perceived to compromise firm performance. This may illustrate that the entrepreneurs in inland China (e.g., Hubei) may still lag those in coastal areas who are more likely to be risk taking and more capable of utilizing their networking capabilities and marketing capabilities to improve performance [85].

Second, when explaining the statistically insignificant impact of dynamic capabilities on international performance, Acosta [31] argued that a probable reason might be the interrelation between international entrepreneurial orientation and international market orientation, since the influence of one factor might cancel the effect of another. Similarly, we believe that the interrelations between international entrepreneurial capabilities and dynamic capabilities might have generated comparable results. Furthermore, entrepreneurial orientation was not considered as a direct determinant of international performance in Matsuno's [85] study.

Finally, consistent with George's [86] arguments, different combinations of dimensions may lead to different conclusions. The insignificant, or even opposite relationship between international entrepreneurship capability and born-global sales performance found here may conform to this pattern.

In addition to the second-order construct results presented in Table 3, we also examine the impact of first-order constructs on the second-order constructs to examine which dimensions can improve a firm's entrepreneurial capabilities to enable born-globals to gain competitive advantages in the global market. The results from Table 4 demonstrate that firms' dynamic capabilities are positively and significantly influenced by adaptation capability, absorption capability and innovation capability. These positive relations imply an increase in these constructs can enable firms to obtain a higher level of dynamic capabilities. Similarly, the results show that international entrepreneurial capability is positively and significantly affected by risk-taking capability, networking capability and marketing capability. The negative relationship between international entrepreneurial capability and sales performance from the second order testing may be explained by the lack of the corresponding capabilities in the first order constructs. In general, the SEM model results show that entrepreneurship can improve firm competitiveness by way of its efforts regarding adaptation, absorption, risk-taking, marketing, innovation, and international networking.

The findings from this research also highlighted the importance of international entrepreneurial capabilities and international knowledge to the sustainable performance of born global firms. In the

process of pursuing sustainable development, firms have to take considerations with their survival and sustainable development in the meantime. The sustainable development is actually highly related with both the realization of business goal and solidification of their market places. The international entrepreneurial capabilities and international knowledge can create sustainable competitive advantages for born globals. For instance, international knowledge especially the preacquisted international knowledge from both entrepreneurs themselves and employees can equipped the firms with prior experiences, and enables the firms to occupy a higher standpoint at the first time and maintain the position in the market for decades.

## 6. Conclusions

This research contributes to research examining the relationship between international entrepreneurship and the performance of Chinese born-globals by taking a more holistic view. Factors such as adaptation capability, absorption capability are considered to enhance international entrepreneurship and relate to sales performance, illustrating the born-globals' competitiveness. Through performing the first and second order SEM analysis, a negative relationship was found between the impact of international entrepreneurial capability on sales performance. We argue that this reveals that entrepreneurs in inland China may still lag entrepreneurs in coastal areas in terms of their risk-taking mindset and behavior. It can also be argued that the interrelationship between international entrepreneurial capability and international marketing capability may account for this result, contradicting with that of previous studies. Nonetheless, we believe that three specific contributions to the literature from this study have been made. First, a broader understanding of the born-global model in the context of China. Previous born-global research has primarily been conducted in developed countries, including the United Kingdom, France, and Canada [87–89]. We also add to the literature addressing Chinese SME performance where specific consideration of born-globals is scarce.

Second, previous research on Chinese born-globals focuses on a specific aspect in each study, such as the impact of international entrepreneurial capability on performancen [32] or the role of leadership [29]. In answering the calls of [30] to investigate the effects other entrepreneurial factors (in addition to international entrepreneurial capability) have on performance, we adopt a holistic approach by incorporating three constructs—dynamic capabilities, international knowledge and international entrepreneurial capability. Furthermore, we propose a way in which the main constructs of international entrepreneurship capability can be further disaggregated and measured by second-order constructs to better understand how international entrepreneurship capability affects sales performance. The use of a second-order structural equation model has enriched the study of international entrepreneurship within China.

Third, differing from the research undertaken by [32], where international knowledge was an indicator of international entrepreneurial capability, the current research treats international knowledge as an independent construct and examines its relationship with born-globals' performance.

## 7. Limitations and Future Research Orientations

The findings have several implications for both entrepreneurs and policy makers. The results show that international knowledge has a significant impact on sales performance. This implies that previous international knowledge may reduce risks associated with operating in international markets, and this knowledge can significantly truncate the preparation period required by a firm. This illuminates the importance of hiring employees who possess international knowledge, or make frequent trips abroad, to establish, or maintain, a good relationship with their clients.

Another implication relates to the importance of international entrepreneurial capabilities. As mentioned, the influence of international entrepreneurship capability takes time to be reflected in firm performance, it is important to let entrepreneurs understand the major components of international entrepreneurial capability and how they may develop it. During their internationalization process,

born-globals may have to prepare for possible short-run negative effects on their performance due to risk-taking in unfamiliar business environments.

Furthermore, during the hard times of Covid-19, small business units world wide all suffered a lot where in Hubei things were getting even worse. The entrepreneurship there from both the global/international side and the recuperative side would no doubt help us identify important research questions that deserve exploration. This can be seen as a best chance to observe small business firms in the epicentre to recover through strengthening their international entrepreneurial capabilities. Since Hubei is an inland province, the small business firms there were more difficult to get financial or fiscal aid from banks and the national financial departments. So in the sense of gaining international knowledge, how to absorb global small business firms' experience and lessons to stimulate members' creativity, use the taxation and financial policies dealing with Covid crisis, maintain and develop international business network might be the optimal plan for them where the possibility of future research locates.

Due to the restricted number of variables and constructs, further studies can attempt to investigate a greater number of variables, such as those relating to the institutional environment, which are believed to have a powerful impact on creating or destroying born global firms in a country. Moreover, because this study is limited in the context of China, the researcher suggests that further study should test the applicability of the framework within the context of other countries. Cultural, political, and economic variables should be included in future studies, and the prospect of a comparative study of born global firms would be a worthwhile endeavor.

The most notable limitation is the relatively small sample size, which may directly affect the generalizability of the results. Many researchers recognize the challenges linked to data collection from SMEs, and this is particularly pertinent regarding SMEs in China [90]. These problems are mainly caused by the lack of centrally compiled SME data in China [91]. The study data were collected from SMEs located in Hubei province. Thus, it would be worthwhile conducting further research by designing a similar, or comparative, study using data from other regions and a larger sample.

Another limitation relates to the restricted number of variables and constructs introduced in the model. Although we combine various constructs that were separately examined in previous research, the constructs or variables used are firm specific or internal factors. Further research may include institutional environment variables, that are believed to have a powerful impact on creating, or destroying, born-globals in a country [92].

The final limitation is that this study used data on firms' exports to distinguish born-globals from those adopting a traditional market entry mode. Foreign direct investment (FDI) may also be a good indicator to capture the feature of born-globals. Therefore, using the level of outward FDI to examine born-globals performance would also be a worthwhile future endeavor.

Based on the limitation mentioned before, the questionnaire was only send out to SMEs located in Hubei province. Thus, it would be worthwhile to conduct further research by designing a similar study that focuses on SMEs in other regions or specific industries. And for the design of questionnaire, it can be improved by adopting dynamic variables such as return on assets (ROA), return on equity (ROE) and so on. These indicators can better reflect the changes in firms performance and measure how the capabilities influence firm's performance in long run. Besides that, further study can improve the design of scales by disperse the responds. In order to avoid the answers being clustered the high end, the questions can be asked from different perspectives, or with reordered scales etc.

**Author Contributions:** Y.Y. developed the theoretical formalism, performed the analytic calculations and performed the numerical simulations. Each author contributed to the final version of the manuscript. All authors have read and agreed to the published version of the manuscript.

**Funding:** This article is supported by "the Fundamental Research Funds for the Central Universities", Zhongnan University of Economics and Law with the fund number: 2722020JCT028.

**Conflicts of Interest:** The authors declare no conflict of interest.

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
