# Peer review of "A Patulous Progress: International Entrepreneurship Effects on Chinese Born-Global Firm Performance"

_sustainability, doi:10.3390/su12145508_

Round 1

Reviewer 1 Report

This a generally well written, informative and accessible paper. The focus is grounded in the literature around dynamic capabilities and international entrepreneurship.

There are opportunities for development in the justification of the empirical context. Whilst the authors provide a rationale for the selection of Hubei province as their focal point for data collection, it is less clear why this region provides a particularly insightful region to focus on SMEs and born globals. A bit more of a regional biography around SME businesses in the region and internationalisation in this area would greatly enhance the empirical justification. This would also address my other area for minor improvement in providing more information on the sample specifically the nature and profile of the SMEs where possible.

The findings, conclusion and limitations sections are clearly written and provide direct and useful insights to the highlighted debates.

Overall I can only see minor improvements around the empirical context that are required.

Author Response

Dear Reviewer, I feel blessed to have such great advices from you. For your first question, the rationale for the selection of Hubei indeed is under questioning. It should be acknowledged that due to the diversity and immensity of local development in China, to identify a standard highly representative sample can be quite difficult. The previous studies on born global firms also collected data from one or two regions in China, so as in this study. But the location of our primary data collection is focused on firms located in Hubei province in central China. When we deciding the selection, we did not mean Hubei is exactly the same as other places in China, but prefer that Hubei is highly representative due to its unique status. Hubei’s total GDP ranked 7th in China with a total GDP of 4582 billion RMB (2019). The annual GDP growth rate in 2019 was 7.5% well above the national average 6.1%. Wuhan, the municipal city of Hubei Province is a special city in China for it is a mixture of multiple missions historically, internationally and economically, it served as the capital of the nation, a heavy industry base, one of the first ports openned to the world with a special court inland but hears the international conflicit cases. The Covid-19 outbreak makes it known to the world. For your advice, this part has been strenghened in line 256-285. For the second question, we specifically gave the detailed description of the firms and managers' information including industry sector, gender, age and educational level. Those information can be found from line 299 to 313 in the revised version attached with this message. All the best!

Reviewer 2 Report

Dear authors,

I find your paper as interesting, with clear and very well delivered introduction and theoretical background and clearly formulated hypotheses however I strongly suggest few improvements:

-at least some further explanation of "Uppsala" should make your point of view more consistent,

-as you mention in chapter 7, "generalizability" could be questionable, therefore I would omit sentences such as "...we believe region´s culture is like the most other areas of China..." (lines 249 & 250), especially in after-COVID situation,

-please specify the time frame WHEN the data were collected - again, some leapfrog to after-COVID recovery/changes would be very beneficial, 

-please specify HOW the likert scale answers were transformed in data inputs for tests performed,

-regarding your regressions, I would strongly suggest to show and describe how you testesd your models for stationarity, heteroscedasticity, and/or other common issues linked to regression models, to prove that presented models are valid,

Best regards...

Author Response

Dear Reviewer,

I feel blessed to have such great advices from you.

For the first question, the description of "Uppsala Model" in detail has been added in line 22 to line 27 of the revised version. In this paper, we assumed that firms which adopt the Uppsala Model seek to establish a solid domestic base at first, and then gradually expand into international markets, with the justification for this relating to the risk and uncertainty embedded in the internationalization process. Therefore, the process of internationalization following the Uppsala model tends to be gradual and incremental.

For the second question, the word generalizability indeed is under questioning and we finally abandoned the sentences such as "...we believe region´s culture is like the most other areas of China...". It should be acknowledged that due to the diversity and immensity of local development in China, to identify a standard highly representative sample can be quite difficult. The previous studies on born global firms also collected data from one or two regions in China, so as in this study. The location of our primary data collection is focused on firms located in Hubei province in central China after careful consideration. When we deciding the generalizability, we did not mean Hubei is exactly the same as other places in China, but prefer that Hubei is highly representative due to its unique status. Hubei’s total GDP ranked 7th in China with a total GDP of 4582 billion RMB (2019). The annual GDP growth rate in 2019 was 7.5% well above the national average 6.1%. Wuhan, the municipal city of Hubei Province is a special city in China for it is a mixture of multiple missions historically, internationally and economically, it served as the capital of the nation, a heavy industry base, one of the first ports opened to the world with a special court inland but hears the international conflict cases. The Covid-19 outbreak makes it known to the world. These points were added to the revision already from line 265 to line 285.

For the third question, the data in this paper was collected from March to July in 2017, an approximate end of China's high speed railway network and the economic geographic landscape of central China did not change since then. Since then the Hubei province can be seen as the absolute central part of China in economic geography and its municipal city Wuhan is the only key node in the Belt & Road Strategy among the several provinces in central China. During the hard times of Covid-19, small business units world wide all suffered a lot where in Hubei things were getting even worse. The entrepreneurship there from both the global/international side and the recuperative side would no doubt help us identify important research questions that deserve exploration. This part can be found in line 297 to line 303.

For the fourth question, the survey participants were asked to respond using the seven‐point Likert scale ranging from 1 (strongly disagree) to 7 (strongly agree), and this point can be found in line 286.

For the final question, due to problems including stationarity, heteroscedasticity, collinearity may existed in cross sectional dataset, and our dataset can be considered as cross sectional, thus we did take these problem into consideration before the fulfillment of structure equation modelling. However, the data we used were collected via questionnaires, and each question is measured as scale, so these problems should not be a major issue in this study.

All the best!

Reviewer 3 Report

The manuscript seems very interesting to me, I congratulate the team for their research.

I only have some recommendations for strengthening it:

In the literature they could include authors such as: Zahra, S. A., & Nambisan, S. (2011). Entrepreneurship in global innovation ecosystems. AMS review, 1 (1), 4 .; Zahra, S. A., Abdelgawad, S. G., & Tsang, E. W. (2011). Emerging multinationals venturing into developed economies: Implications for learning, unlearning, and entrepreneurial capability. Journal of Management Inquiry, 20 (3), 323-330 .; and Nambisan, S., Zahra, S. A., & Luo, Y. (2019). Global platforms and ecosystems: Implications for international business theories. Journal of International Business Studies, 50 (9), 1464-1486. Studies that could contribute to the development of the literature on international entrepreneurship and dynamic capacities.

In the methodology section, it is convenient to show in detail the main characteristics of the companies such as gender, age, and educational level of the manager. Furthermore, it is important to show the sector of activity of the companies participating in the study.

In the results section: It is recommended to show the results of the discriminant validity of the model (through a table). Regarding the hypotheses, it is recommended to add the number in each structural relationship and show it in the tables that have been put in the manuscript.

In the results section, we recommend adding the theoretical implications in greater detail (do the findings align with the theory of dynamic capabilities with the theory of resources and capabilities?). As well as explaining with greater precision and detail the empirical implications that the study has generated for the business sector, such as: what are the barriers, problems, and challenges that managers of these companies have or face to have a greater projection in international markets and take risks? In summary, explain what managers need to do to improve dynamic capacities (adaptation, innovation, and absorption) and thus be able to fully exploit these capacities to be considered as a higher-level capability.

Add future lines of research and consider studies with mediating or moderating variables in order to support this type of study.

Author Response

Dear Reviewer,   I feel blessed to have such great advices from you.   For the first advice, we have added works of authors you mentioned into the newest version in line 43 to 48 as insightful fulfillment of the literature review.   For the second advice, we specifically gave the detailed description of the managers' information including industry sector, gender, age and educational level. The general information about the managers can be found in line 299 to line 313.   For the third advice, the results of the discriminant validity of the model can be found in table 2 (line 381).   For the fourth advice, we indeed adding the theoretical implications in greater detail from line 528 to line 534 and offered a plan for managers about actions needed when faced with risks in line 520 to 527.   For the final advice, we considered studies with mediating or moderating variables in order to support our paper and other similar ones through optimizing our questionnare and test other places both in China and around the world. This part can be found in line 550 to 557.   All the best!

Round 2

Reviewer 2 Report

Dear authors,

Thank you for your comments and efforts shown in your paper after the review. 

Best regards...